analytical chemistry/green chemistry

salting-out liquid–liquid extraction, HPLC-DAD, vitamin D3, milk samples

**Author for correspondence:**
Mazidatulakmam Miskam
e-mail: mazidatul@usm.my

# Salting-out assisted liquid–liquid extraction coupled with high-performance liquid chromatography for the determination of vitamin D3 in milk samples

Nur Hidayah Sazali[1], Anas Alshishani[2], Bahruddin Saad[3], Ker Yin Chew[4], Moi Me Chong[4] and Mazidatulakmam Miskam[1]

[1]School of Chemical Sciences, Universiti Sains Malaysia, 11800 Pulau Pinang, Malaysia
[2]Faculty of Pharmacy, Zarqa University, 13132 Zarqa, Jordan
[3]Fundamental and Applied Sciences Department, Universiti Teknologi PETRONAS, 32610 Seri Iskandar, Perak, Malaysia
[4]MYCO2 Laboratory Sdn Bhd, Lengkok Kikik 1, Taman Inderawasih, 13600 Pulau Pinang, Malaysia

MM, 0000-0001-9757-0883

In this study, salting-out assisted liquid–liquid extraction (SALLE) as a simple and efficient extraction technique followed by high-performance liquid chromatography (HPLC) was employed for the determination of vitamin D3 in milk samples. The sample treatment is based on the use of water-miscible acetonitrile as the extractant and acetonitrile phase separation under high-salt conditions. Under the optimum conditions, acetonitrile and ammonium sulfate were used as the extraction solvent and salting-out agent, respectively. The vitamin D3 extract was separated using Hypersil ODS (250x i.d 4.6 mm, 5 µm) HPLC column that was coupled with diode array detector. Vitamin D2 was used as internal standard (IS) to offset any variations in chromatographic conditions. The vitamin D3 and the IS were eluted in 18 min. Good linearity ($r^2 > 0.99$) was obtained within the range of 25–600 ng g$^{-1}$ with the limit of detection of 15 ng g$^{-1}$ and limit of quantification of 25 ng g$^{-1}$. The validated method was applied for the determination of vitamin D3 in milk samples. The recoveries for spiked samples were from 94.4 to 113.5%.

# 1. Introduction

Cholecalciferol, commonly known as vitamin D3, is a fat-soluble vitamin, which is of great nutritional interest, that supports metabolism processes and improves the efficiency of proteins and enzymes [1]. It plays important roles within the body as it regulates blood calcium and phosphorus levels by promoting the absorption in intestines and reabsorption of calcium in the kidney that is key to the mineralization of the bones [2,3]. Vitamin D deficiency is recognized as one of the most common mild chronic medical conditions in the world. It can lead to soft, thin and brittle bones, a disease known as rickets in children [4,5].

Vitamin D3 is naturally found in trace amounts in some foods such as some fatty fishes (mackerel, salmon and sardines), fish liver oils and eggs from hens that have been fed vitamin D3. Among dairy products, processed milk products and infant formulas are fortified with vitamin D3. However, its content is low, ranging from 4 to 40 IU (0.1–1.0 µg l$^{-1}$) and which corresponds mainly to vitamin D3 and 25-hydroxy-vitamin D3 [6]. Vitamin D3 is also sensitive to heat and light and is easily oxidized [7].

Sample preparation process plays an important role in enhancing sensitivity and reducing matrix interference. The most common preparation technique is saponification [8,9], liquid–liquid extraction (LLE) [10], solid-phase extraction (SPE) [11–14] and derivatization [15]. Saponification is commonly the first step before LLE and SPE and is used to remove neutral lipids, especially triglycerides. This method has been used for the determination of vitamin D in various types of food samples such as infant products [12,16], milk, cream and butter [17], human foods, pet foods and supplements [18], fish oil [19], beef and poultry [20], fortified orange juice [21], dietary supplements and vitamin premixes [22], bovine milk [11], vegetables [23], meat [24] and human serum [6]. Saponification using methanolic or ethanolic KOH at elevated temperatures (60–100°C) for 20–45 min was able to suppress lipids and proteins interferences [25]. A disadvantage of the heat treatment is the possibility of isomerization of vitamin D3 and this causes difficulty in liquid chromatography (LC) analysis.

LLE and SPE involve high consumption of solvents during the multi-step extraction and the long-time sample preparation. Research in vitamin D3 chemistry needs to be supported by the development of rapid and sensitive modern analytical methods and efficient sample clean-up procedures. Quantification of vitamin D3 in dairy products has been considered one of the most difficult goals because of its low content in complex matrix [25].

Salting-out assisted liquid–liquid extraction (SALLE) is another interesting approach for extraction of vitamin D3. Salting-out is a process of addition of electrolytes to an aqueous phase to increase the distribution of a solute. SALLE can be viewed as modification to the conventional LLE procedure so that the extract is compatible for LC analysis. In the SALLE procedure, water-miscible organic solvents such as acetonitrile (ACN) and salts such as sodium chloride or ammonium sulfate are added. After gentle mixing and centrifugation, the organic solvent is 'salted-out' and formed a separate phase on the top of the aqueous phase. The extract can be conveniently sampled and injected directly into LC for analysis.

When this sample pretreatment method was applied for the determination of aflatoxins in milk samples, the analytes were not only extracted but proteins were precipitated, separated the fats from the water-soluble components (lactose and minerals) [26]. It is noted that most applications reported with SALLE method are focused on the extraction of polar compounds [27,28].

It is clear from the above discussions that alternative sample pretreatments that are simpler but maintaining the integrity of the vitamins D2 and D3 are required. Efforts should be directed towards eliminating the saponification step and performing at room temperature to prevent isomerism and decomposition. Towards this end, of the plethora of microextraction techniques, the SALLE seemed to be the best candidate. Thus, this paper is dedicated to the development of a SALLE pretreatment method for the determination of vitamin D3 in milk.

# 2. Experimental

## 2.1. Chemicals and reagents

The analytical standard of vitamin D3, high-performance liquid chromatography (HPLC)-grade methanol and acetonitrile were purchased from Merck (Darmstadt, Germany). Vitamin D2 and vitamin C (L-ascorbic acid) were obtained from Sigma-Aldrich (St Louis, MO, USA). Analytical grade ethanol (denatured) and ammonium sulfate were purchased from QReC (Auckland, New Zealand). Formic acid (99%) was supplied by Univar (Auburn, Australia). Distilled water (resistivity, 18.2 mΩ cm$^{-1}$) was

produced by Milipore water purification system (Molsheim, France) and was used throughout for the preparation of solutions. Vitamin D3 and D2 stock solutions (1000 ng g$^{-1}$) in methanol were used to prepare standard solutions.

## 2.2. Instrumentation

HPLC Perkin Elmer Series 200 (Waltham, MA, USA) with diode-array detector (DAD) were used. Separation was performed on Hypersil ODS (250x i.d 4.6 mm, 5 µm) from Thermo Fisher Scientific (Waltham, MA, USA). Methanol : water (98 : 2 (v/v) %) as mobile phase at flow rate 0.8 ml min$^{-1}$ were used. The acquisition of data was done at 265 nm for both vitamins (D3 and D2). The injection volume used for the determination was 100 µl and both vitamins were eluted within 18 min. The process was controlled by TotalChrom™ software.

## 2.3. Vitamin D3 stability tests

Preliminary experiments were conducted to test the stability of vitamin D3 after exposure to external factors such as light, elevated temperature and hydrogen peroxide (oxidizing agent). The standard vitamin D3 was stored in different storage conditions: 25°C or 40°C exposed to light, and with addition of hydrogen peroxide (2–10 v/v %). Immediately after collection, the standards from each condition were analysed using HPLC. The stability of vitamin D3 under these conditions was determined by calculating the percentage change in concentration relative to fresh sample.

## 2.4. SALLE procedure

The milk samples were obtained from local stores in Gelugor, Penang, Malaysia. Milk samples (5 g) were weighed and dissolved in distilled water (10 ml) at room temperature. The mixtures were sonicated for 30 min in order to ensure that the milk samples were completely dissolved. In order to prevent analyte loss due to the light and temperature effects, the mixtures were covered with aluminium foils and sonicated at room temperature. After 30 min sonication, ACN (3 ml) was added followed by the addition of vitamin C in ethanol [16] into the mixtures (150 µl). Next, the mixtures were sonicated (60 min) and mixed for 10 min. About 7 g of ammonium sulfate ($(NH_4)_2SO_4$) and formic acid (50 µl) were added into the mixtures. The mixtures were then mixed well until the salt was completely dissolved before centrifuging (8500 r.p.m.) for 30 min. The upper layer (about 1 ml) was transferred into small vials and evaporated using $N_2$ stream. The residues were reconstituted using a mobile phase (150 µl) before injected into the HPLC system.

## 2.5. Method validation

The method was validated for linearity, detection and quantification limits, repeatability and recoveries. The linearity of the method was assessed by plotting a calibration curve over 25–600 ng g$^{-1}$ of standard vitamin D3. The detection and quantification limits were determined from the slope of calibration curve. For a linear calibration curve, it can be expressed by equation, $y = a + bx$. In this model, the sensitivity, $b$ and the limit of detection (LOD) can be determined as

$$D = 3 \times \frac{\text{s.d.}}{b}, \quad \text{while} \quad \text{LOQ} = 10 \times \frac{\text{s.d.}}{b}.$$

The determination of recovery was performed by spiking three different concentrations of standard vitamin D3 in milk samples. The recovery was calculated based on the measured vitamin D3 concentration data as shown below:

$$\text{recovery (\%)} = \frac{\text{concentration of spiked sample} - \text{concentration of sample}}{\text{concentration of standard}} \times 100.$$

The precision of the method was evaluated in terms of repeatability (intra-day precision) and intermediate precision (inter-day precision). Experiments were carried out for milk samples at three different concentration levels of vitamin D3 (50, 200 and 600 ng g$^{-1}$). Repeatability was evaluated for six samples prepared and injected in triplicate on the same day, under the optimum conditions. Intermediate precision was evaluated using a similar procedure, but the samples were analysed on

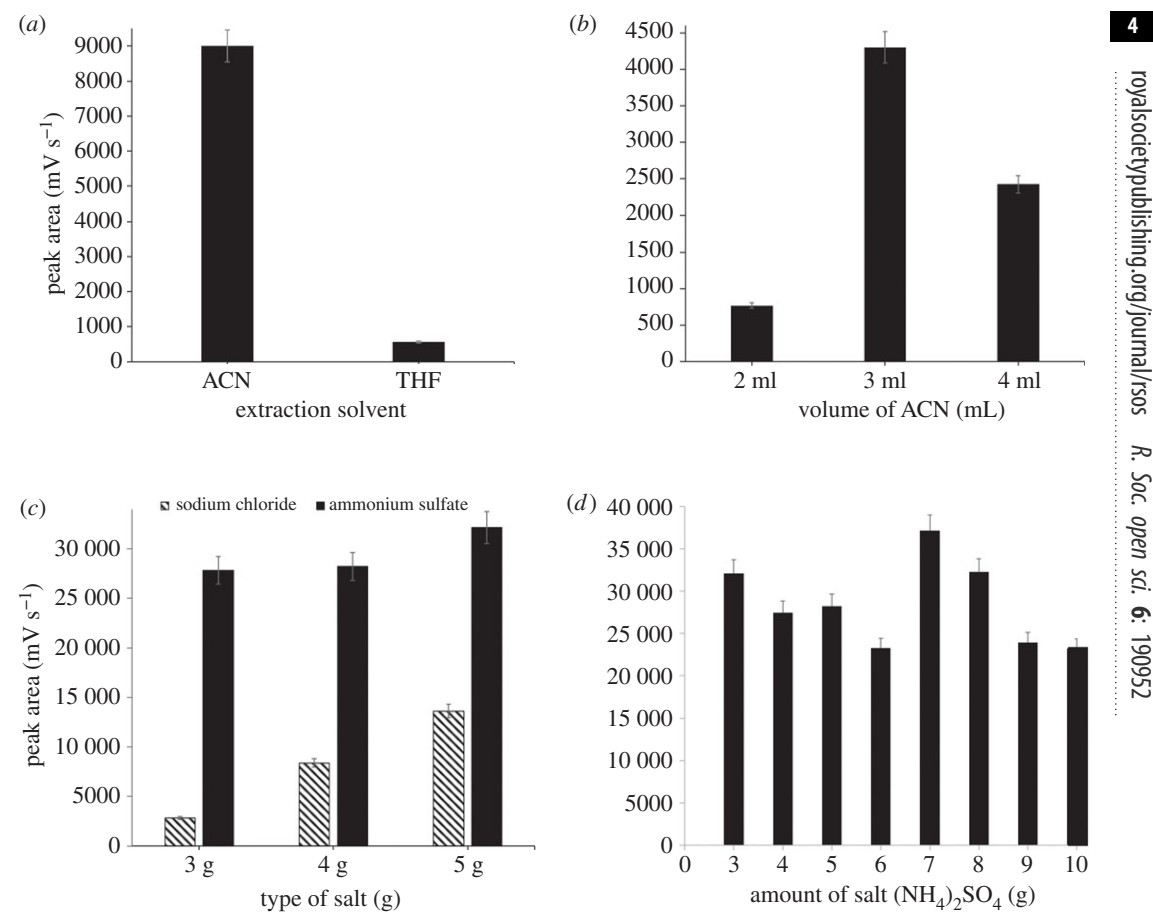

**Figure 1.** Optimization of salting out liquid–liquid extraction parameters (*a*) type of extraction solvents; (*b*) volume of ACN; (*c*) type of salt; (*d*) amount of salt. Extraction condition: sample volume, 10 ml; time of sonication, 60 min.

three consecutive days. The milk powders were spiked with three different levels of concentrations; 50, 200 and 600 ng ml$^{-1}$ for recovery studies.

# 3. Result and discussion

## 3.1. Preliminary studies of vitamin D3 stability

Vitamin D3 is known to degrade under various conditions; thereby its analysis is considered challenging. Preliminary tests were conducted to test the stability of vitamin D3 after exposure to the light, elevated temperature and oxidizing agent under the present experimental conditions. These tests were necessary to provide information on the handling of reagents prior to the SALLE procedure. About 65% of vitamin D3 was degraded after being exposed to these conditions, in agreement with earlier studies [29]. Thus, it was decided that an antioxidant (ascorbic acid) would be used during the SALLE procedure.

## 3.2. Effect of type of extraction solvent

The type of extraction solvent is an important parameter in SALLE procedure. The ideal extraction solvent should be miscible in water, polar and able to induce phase separation upon addition of salt [27]. In this study, tetrahydrofuran (THF) and acetonitrile (ACN) were tested as extraction solvents. ACN gave better results compared to THF for the extraction of vitamin D3 as shown in figure 1*a*. ACN is a highly recommended solvent because it has high polarity and is miscible in water. Upon the addition of salts, ACN is able to precipitate proteins and caused partition of the analyte into organic solvent. Besides, ACN is a less harmful organic solvent compared to THF.

**Table 1.** Recovery and precision obtained for different spiked concentrations of vitamin D3.

| concentration spiked (ng g$^{-1}$) | recovery % (average ± s.d. %) | precision (% r.s.d.) |
|---|---|---|
| 50 | 94.40 ± 10.8 | 11.4 |
| 200 | 113.5 ± 1.78 | 1.56 |
| 600 | 106.6 ± 4.34 | 4.07 |

## 3.3. Effect of volume of extraction solvent

The mass transfer of the analytes is expected to increase as the volume of ACN is increased due to the enhancement of partitioning of analytes between the two phases. Different volumes of the extraction solvent (2–4 ml) were investigated (figure 1$b$). It was found that the peak areas gradually increased as the volume of ACN increased up to 3 ml and decreased afterwards due to dilution factor effect. This is because at a lower volume (2 ml), the separated organic phase was not obvious, and the collection of the organic phase was difficult. Thus, 3.0 ml of ACN was selected for the extraction of vitamin D3 from milk samples.

## 3.4. Effect of addition of salts

The type of salting-out reagent is an important consideration in the SALLE procedure. Different type and amount of salts affect the degrees of phase separation. In this study, the effects of sodium chloride and ammonium sulfate were evaluated. The results demonstrated that ammonium sulfate provided better phase separation and higher peak areas compared to sodium chloride (figure 1$c$). Ammonium sulfate ($(NH_4)_2SO_4$) is often used as salting-out reagent because of its high solubility, allowing for solutions to be prepared at high ionic strength, low price and its easy availability [30]. Thus, $(NH_4)_2SO_4$ was selected for the subsequent study. The effect of different amounts of $(NH_4)_2SO_4$ (3.0–9.0 g) on the extraction efficiency was also investigated. The optimum amount of salt was 7.0 g (figure 1$d$). The peak area of the extracted vitamin D3 increased with increasing amount of salt up to 7.0 g and the area decreased thereafter.

## 3.5. Effect of sonication time

Sonication treatment has two main purposes, i.e. to improve the extraction yields and to accelerate the phase separation. After the addition of ACN into the sample solution, the effect of sonication was examined. The extraction yields of vitamin D3 obtained at different times of sonication (30 and 60 min) showed an increase in peak area as a function of time. Sonication treatment for 60 min provides good extraction of vitamin D3 from the aqueous phase into the ACN phase. Sonication treatment more than that should be avoided, because the temperature of the sample solution started to increase, which could cause analyte loss due to excessive heating, as the stability of vitamin D3 was affected by temperature at prolonged time. For subsequent analysis, 60 min of sonication was chosen.

## 3.6. Adopted extraction conditions

The adopted extraction conditions for 10 ml sample were performed with 3 ml ACN as the extraction solvent and 7.0 g of $(NH_4)_2SO_4$ salt as the salting-out reagent with 60 min of sonication.

## 3.7. Method validation

Under the above optimum experimental conditions, the proposed SALLE method was validated in terms of linearity, LOD, limits of quantification (LOQ), repeatability and recovery. The calibration curve was obtained by linear regression of peak area ratio of D3 over internal standard (IS), versus concentration of vitamin D3 (25–600 ng g$^{-1}$). Typical plot is described by

$$y = 3.4634x - 0.0215,$$

where $y$ is the ratio peak area of vitamin D3 and IS and $x$ is the concentration of vitamin D3 in ng g$^{-1}$. The coefficient of determination, $r^2$ obtained is about 0.99 indicating the excellent linearity between ratio of

**Table 2.** Levels (ng g$^{-1}$) of vitamin D3 in milk samples.

| sample | proposed method (ng g$^{-1}$) (average ± s.d.) | recovery (%) | r.s.d. (%) |
|---|---|---|---|
| milk powder 1 | 43.4 ± 0.03 | 102.5 | 6.5 |
| milk powder 2 | 69.0 ± 0.001 | 73.9 | 17.9 |
| milk powder 3 | 48.0 ± 0.005 | 95.4 | 9.9 |
| milk powder 4 | 48.1 ± 0.004 | 96.2 | 8.5 |
| liquid milk 1 | 38.8 ± 0.004 | 64.5 | 9.9 |
| liquid milk 2 | 21.6 ± 0.001 | 86.0 | 4.1 |
| liquid milk 3 | 20.2 ± 0.002 | 80.4 | 8.0 |
| liquid milk 4 | 18.1 ± 0.001 | 69.6 | 2.8 |

**Table 3.** Comparison between the proposed SALLE-HPLC method for the determination of vitamin D3 in milk samples and some reported methods.

| samples | extraction technique | extraction time (min) | technique | LOQ (ng g$^{-1}$) | recovery (%) | ref |
|---|---|---|---|---|---|---|
| milk product (solid, liquid) | SALLE | 60 min (without saponification) | HPLC-DAD | 25 | 94.4–113.5 | this work |
| milk product (solid, liquid) | LLE | 70 min saponification + extraction time | LC-UV | 100 | 93.0–102.0 | [16] |
| infant formula | DLLME | — | LC-MS | 1 | 88.0–103.0 | [23] |
| infant formula | LLE dSPE | 10 min saponification + extraction time | LC-MSMS | 1 | 93.1–110.6 | [12] |
| infant formula | LLE | overnight saponification + 10 min extraction time | HPLC-MSMS | 5 | 89.5–115.0 | [18] |
| milk product (solid) | LLE | overnight saponification + 20 min extraction time | HPLC-DAD | 20 | 73.0–115.0 | [31] |
| milk product (solid) multi-vitamin | packed fibre SPE | — | HPLC-DAD | 70 | 91.7–104.5 | [22] |
| cod liver fish oil supplement | SPE | — | LC-MS | 0.1 | 88.8–99.3 | [19] |

peak areas of vitamin D3 and IS versus concentration of vitamin D3. The LOD and LOQ obtained were 15 and 25 ng g$^{-1}$, respectively. The LOD and LOQ obtained using this method are about similar with previous report using saponification and LLE [31]. This suggests that the SALLE method may be used to analyse milk at ng g$^{-1}$ levels without saponification.

Recovery was evaluated by spiking the milk samples with standard vitamin D3 at different concentrations (table 1). The recoveries were acceptable (94.4–113.5%). The precision of the analytical

procedure was determined using six replicates samples. The repeatability obtained was good (below 15%). It was found that the precision (% r.s.d.) for three concentration levels (50, 200 and 600 ng g$^{-1}$) was less than 11.5%.

## 3.8. Analysis of milk samples

In order to evaluate the applicability of the proposed method, different milk samples were analysed and results were compared with the amount claimed by manufacturers (table 2). All analysed samples contained vitamin D3 concentrations that were well as reported by the manufacturers. The average tested values for milk samples were in range 18.1–69.0 ng g$^{-1}$. The recoveries obtained ranged from 64.5 to 102.5%, and the relative standard deviation (r.s.d.) values ranged from 2.8 to 17.9%.

A student paired *t*-test (at 95% confidence level) shows that there is any significant difference in the mean values of the manufacturer's claimed values and that of the proposed method (*t* stat, 0.08; *t* critical, 2.2).

## 3.9. Comparison to previously reported methods

Table 3 summarizes the important characteristics of proposed method for the determination of vitamin D3 in milk samples compared to some previously reported methods. It is readily apparent that the developed method using SALLE has many advantageous features especially in terms of solvent consumption, sample preparation time and sensitivity. This proposed method is simple, since there is no saponification method involved, unlike some previously reported methods. Some of the previously reported methods use SPE, but this requires multi-steps (equilibration, washing, loading and elution) with single-use SPE cartridges, which makes the experiment longer.

# 4. Conclusion

Significant advancement is sample pretreatment method based on SALLE for the HPLC determination of vitamin D3 in milk is demonstrated. This method is simple, fast and consumes less amount of solvent compared to previous sample pretreatment methods which require saponification and elevated temperatures. The extracts are readily injected for the HPLC analysis, unlike previous methods that require long and gentle evaporation to remove extracting solvents. It is indeed very interesting if the SALLE technique can be adopted, with minor modifications, to be applied to the analysis of other physiologically important metabolites of vitamin D such as 25-OH-D. If this is possible, it will be a major breakthrough in the analysis of vitamin D metabolites.

Data accessibility. Data are available from the Dryad Digital Repository: https://doi.org/10.5061/dryad.cq3bp7m [32].
Authors' contributions. K.Y.C., M.M.C., A.A., B.S. and M.M. participated in the design of the study; N.H.S., K.Y.C. and M.M.C. carried out most of the laboratory work, accomplished the data analysis and drafted the manuscript; M.M., A.A. and B.S. conceived the study, coordinated the study and helped draft the manuscript. All authors gave final approval for publication.
Competing interests. The authors declare that they have no competing interests.
Funding. This work was supported by funding from Universiti Sains Malaysia Research Grants (Short Term: 304.PKIMIA.6313334; Bridging: 304.PKIMIA.6316492).
Acknowledgements. The authors thank the editors and anonymous reviewers for their helpful suggestions for this manuscript.

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
