## [Reviewer comments · Royal Society Open Science]

Review History

RSOS-182170.R0 (Original submission)

Review form: Reviewer 1

Is the manuscript scientifically sound in its present form?

Yes

Are the interpretations and conclusions justified by the results?

Yes

Is the language acceptable?

No

Is it clear how to access all supporting data?

Yes

Do you have any ethical concerns with this paper?

No

Have you any concerns about statistical analyses in this paper?

No

Recommendation?

Reject

Comments to the Author(s)

Dear authors,

I have read with much interest the manuscript entitled: "Salting-out assisted liquid-liquid extraction coupled with high-performance liquid chromatography for determination of vitamin D3 in milk samples. But I recommend to improve your paper, in particular:

1. The introduction section is too general. For instance, there is no needs to mention in details in which samples vitamin D3 was found. It will be enough to write generally (for e.g. food samples, dietary supplements and biological samples, etc.). But information about SALLE should be included, its principles and benefits. Please, transfer information about SALLE from Result and Discussion section to Introduction one.
2. It is not clear how the conditions of HPLC separation were selected. Was performed any optimization? Information about injection volume, run time, software should be provided.
3. Line 45 p. 4. The word "extraction" is repeated 2 times . I would recommend to change to "SALLE procedure" or "Extraction procedure".
4. The description of extraction procedure is not clear. How was performed experiment? What tools were used?
5. Line 59, p. 4. It is not clear how much of vitamin C was added. Why was it necessary to prepare vitamin C in ethanol?
6. Line 6, p. 5. Why was added formic acid?
7. In Result and Discussion section should be provided only obtained results. Line 22-46, there is known information, which can be transferred to introduction section.
8. Preliminary study is not clear. The results of this study are not provided, just only mention the reference.
9. Line 15, 38 p.7. The titles of these items can be change to "Effect of type of extraction solvent", "Effect of extraction solvent volume".
10. Line 17-33 p. 7. A lot of a well-known information, but this is not the results of this experiment. Here should be provided only the results of the test.
11. Line 26, p. 7. At the beginning of Results and discussion section it was mention about magnesium sulfate, ammonium acetate (but, not only these salts can be used for phase separation), after the effect of NaCl and ammonium sulfate was studied (line 6, p 8).
12. The coefficient of determination should contain 4 digits (line 39, p.9).
13. Please provide recovery and RSD to Table 2. From my point of view, the developed method is not suitable for analysis milk powder 1 and 3 (from your data, RSD in this sample is about 81%), for milk powder 3 RSD is 105.5%). How can you explain these results of analysis? What did effect on such critical deviation? In Table 2, I would to recommend to write results in ng/g. The way of writing of SD as 0.000 is not acceptable (liquid milk 3, liquid milk 4).
14. In section Comparison to previously reported methods, referenced should be added and in Table 2 references should be modified according to the same references style like in the whole text.
15. To the capture of Fig 1 should be added information about extraction conditions. Fig 3b,c (ml and g should be mention only one time (on the scale or in the title of the axis).

Review form: Reviewer 2

Is the manuscript scientifically sound in its present form?

Yes

Are the interpretations and conclusions justified by the results?

Yes

Is the language acceptable?

Yes

Is it clear how to access all supporting data?

Yes

Do you have any ethical concerns with this paper?

No

Have you any concerns about statistical analyses in this paper?

No

Recommendation?

Accept with minor revision (please list in comments)

Comments to the Author(s)

The paper is written in a concise and well understood way and fit with the scope of Royal Society Open Science. In the chapter Introduction, the authors of the work in a clear way introduced into the subject of work and presented current knowledge on the subject. The aim of the work was precisely defined. The Experimental chapter do not raise any objections. The discussion of the results was carried out correctly. As well as formulated conclusions are correct.

There are only a series of comments that have been introduced in the manuscript. All remarks are given in the text (Appendix A).

Decision letter (RSOS-182170.R0)

12-Mar-2019

Dear Dr Miskam:

Manuscript ID: RSOS-182170

Title: "Salting-out assisted liquid-liquid extraction coupled with high performance liquid chromatography for determination of vitamin D3 in milk samples"

Thank you for submitting the above manuscript to Royal Society Open Science. Your paper was sent to reviewers and their comments are included at the bottom of this letter.

In view of the concerns raised by the reviewers, the manuscript has been rejected in its current form. However, a new manuscript may be submitted which takes into consideration these comments.

Please note that resubmitting your manuscript does not guarantee eventual acceptance, and that your resubmission will be subject to peer review before a decision is made.

Your resubmitted manuscript should be submitted by 09-Sep-2019. If you are unable to submit by this date please contact the Editorial Office.

On behalf of the Subject Editor Professor Anthony Stace and the Associate Editor Dr Ya-Wen Wang

REVIEWER(S) REPORTS:
Associate Editor Comments to Author ():
RSC Associate Editor:
Comments to the Author:
(There are no comments.)

RSC Subject Editor:
Comments to the Author:
(There are no comments.)

Reviewers' Comments to Author:
Reviewer: 1

Comments to the Author(s)

Dear authors,

I have read with much interest the manuscript entitled: "Salting-out assisted liquid-liquid extraction coupled with high-performance liquid chromatography for determination of vitamin D3 in milk samples. But I recommend to improve your paper, in particular:

1. The introduction section is too general. For instance, there is no needs to mention in details in which samples vitamin D3 was found. It will be enough to write generally (for e.g. food samples, dietary supplements and biological samples, etc.). But information about SALLE should be

included, its principles and benefits. Please, transfer information about SALLE from Result and Discussion section to Introduction one.

2. It is not clear how the conditions of HPLC separation were selected. Was performed any optimization? Information about injection volume, run time, software should be provided.
3. Line 45 p. 4. The word "extraction" is repeated 2 times . I would recommend to change to "SALLE procedure" or "Extraction procedure".
4. The description of extraction procedure is not clear. How was performed experiment? What tools were used?
5. Line 59, p. 4. It is not clear how much of vitamin C was added. Why was it necessary to prepare vitamin C in ethanol?
6. Line 6, p. 5. Why was added formic acid?
7. In Result and Discussion section should be provided only obtained results. Line 22-46, there is known information, which can be transferred to introduction section.
8. Preliminary study is not clear. The results of this study are not provided, just only mention the reference.
9. Line 15, 38 p.7. The titles of these items can be change to "Effect of type of extraction solvent", "Effect of extraction solvent volume".
10. Line 17-33 p. 7. A lot of a well-known information, but this is not the results of this experiment. Here should be provided only the results of the test.
11. Line 26, p. 7. At the beginning of Results and discussion section it was mention about magnesium sulfate, ammonium acetate (but, not only these salts can be used for phase separation), after the effect of NaCl and ammonium sulfate was studied (line 6, p 8).
12. The coefficient of determination should contain 4 digits (line 39, p.9).
13. Please provide recovery and RSD to Table 2. From my point of view, the developed method is not suitable for analysis milk powder 1 and 3 (from your data, RSD in this sample is about 81%, for milk powder 3 RSD is 105.5%). How can you explain these results of analysis? What did effect on such critical deviation? In Table 2, I would to recommend to write results in ng/g. The way of writing of SD as 0.000 is not acceptable (liquid milk 3, liquid milk 4).
14. In section Comparison to previously reported methods, referenced should be added and in Table 2 references should be modified according to the same references style like in the whole text.
15. To the capture of Fig 1 should be added information about extraction conditions. Fig 3b,c (ml and g should be mention only one time (on the scale or in the title of the axis).

Reviewer: 2

Comments to the Author(s)

The paper is written in a concise and well understood way and fit with the scope of Royal Society Open Science. In the chapter Introduction, the authors of the work in a clear way introduced into the subject of work and presented current knowledge on the subject. The aim of the work was precisely defined. The Experimental chapter do not raise any objections. The discussion of the results was carried out correctly. As well as formulated conclusions are correct.

There are only a series of comments that have been introduced in the manuscript. All remarks are given in the text.

Author's Response to Decision Letter for (RSOS-182170.R0)

See Appendices B & C.

RSOS-190952.R0

Review form: Reviewer 1

Is the manuscript scientifically sound in its present form?

Yes

Are the interpretations and conclusions justified by the results?

Yes

Is the language acceptable?

Yes

Do you have any ethical concerns with this paper?

No

Have you any concerns about statistical analyses in this paper?

No

Recommendation?

Accept with minor revision (please list in comments)

Comments to the Author(s)

It would be good to make some changes in the manuscript:

1. page 5, line 58, Please add reference. From the method description, it is not clear, what concentration of Vitamin C was.
2. Avoid the excess of figure numbers in all the parts of the manuscript and, specially in the tables (for e.g. 94.4+-10.8, 113.5+-1.8).

Review form: Reviewer 2

Is the manuscript scientifically sound in its present form?

Yes

Are the interpretations and conclusions justified by the results?

Yes

Is the language acceptable?

Yes

Do you have any ethical concerns with this paper?

No

Have you any concerns about statistical analyses in this paper?

No

Recommendation?

Accept as is

Comments to the Author(s)

I would like thank the authors their interest in improving the article

Decision letter (RSOS-190952.R0)

08-Jul-2019

Dear Dr Miskam:

Title: Salting-out assisted liquid-liquid extraction coupled with high performance liquid chromatography for determination of vitamin D3 in milk samples

Manuscript ID: RSOS-190952

Thank you for submitting the above manuscript to Royal Society Open Science. On behalf of the Editors and the Royal Society of Chemistry, I am pleased to inform you that your manuscript will be accepted for publication in Royal Society Open Science subject to minor revision in accordance with the referee suggestions. Please find the reviewers' comments at the end of this email.

The reviewers and handling editors have recommended publication, but also suggest some minor revisions to your manuscript. Therefore, I invite you to respond to the comments and revise your manuscript.

Because the schedule for publication is very tight, it is a condition of publication that you submit the revised version of your manuscript before 17-Jul-2019. Please note that the revision deadline will expire at 00.00am on this date. If you do not think you will be able to meet this date please let me know immediately.

- 1) A text file of the manuscript (tex, txt, rtf, docx or doc), references, tables (including captions) and figure captions. Do not upload a PDF as your "Main Document".
- 2) A separate electronic file of each figure (EPS or print-quality PDF preferred (either format should be produced directly from original creation package), or original software format)
- 3) Included a 100 word media summary of your paper when requested at submission. Please ensure you have entered correct contact details (email, institution and telephone) in your user account
- 4) Included the raw data to support the claims made in your paper. You can either include your data as electronic supplementary material or upload to a repository and include the relevant doi within your manuscript

5) All supplementary materials accompanying an accepted article will be treated as in their final form. Note that the Royal Society will neither edit nor typeset supplementary material and it will be hosted as provided. Please ensure that the supplementary material includes the paper details where possible (authors, article title, journal name).

Best wishes,

Dr Laura Smith
Publishing Editor, Journals

RSC Associate Editor
Comments to the Author:
(There are no comments.)

Reviewer comments to Author:
Reviewer: 2

Comments to the Author(s)
I would like thank the authors their interest in improving the article

Reviewer: 1

Comments to the Author(s)
It would be good to make some changes in the manuscript:
1. page 5, line 58, Please add reference. From the method description, it is not clear, what concentration of Vitamin C was.
2. Avoid the excess of figure numbers in all the parts of the manuscript and, specially in the tables (for e.g. 94.4+-10.8, 113.5+-1.8).

Author's Response to Decision Letter for (RSOS-190952.R0)

See Appendix D.

Decision letter (RSOS-190952.R1)

30-Jul-2019

Dear Dr Miskam:

Title: Salting-out assisted liquid-liquid extraction coupled with high performance liquid chromatography for the determination of vitamin D3 in milk samples
Manuscript ID: RSOS-190952.R1

It is a pleasure to accept your manuscript in its current form for publication in Royal Society Open Science. The chemistry content of Royal Society Open Science is published in collaboration with the Royal Society of Chemistry.

RSC Associate Editor
Comments to the Author:
(There are no comments.)

Reviewer(s)' Comments to Author:

Appendix A**ROYAL SOCIETY
OPEN SCIENCE****Salting-out assisted liquid-liquid extraction coupled with
high performance liquid chromatography for determination
of vitamin D3 in milk samples**

Journal:	Royal Society Open Science
Manuscript ID	RSOS-182170
Article Type:	Research
Date Submitted by the Author:	10-Feb-2019
Complete List of Authors:	Sazali, Nur Hidayah; Universiti Sains Malaysia, School of Chemical Sciences Al-shishani, Anas Saad, Bahruddin; Universiti Teknologi PETRONAS, Fundamental & Applied Sciences Chong, Moi Me; MYCO2 Laboratory Sdn Bhd Chew, Ker Yin; MYCO2 Laboratory Sdn Bhd Miskam, Mazidatulakman; Universiti Sains Malaysia, School of Chemical Sciences
Subject:	Analytical chemistry < CHEMISTRY, Green chemistry < CHEMISTRY
Keywords:	Salting-out assisted liquid-liquid extraction, vitamin D3, High performance liquid chromatography, Milk samples
Subject Category:	Chemistry

**Salting-out assisted liquid-liquid extraction coupled with high performance liquid**
**chromatography for determination of vitamin D3 in milk samples**

Nur Hidayah SAZALI^a, Anas ALSHISHANI^a, Bahruddin SAAD^b, Ker Yin CHEW^c, Moi Me
CHONG^c, Mazidatulakmam MISKAM^{*a}

*^a School of Chemical Sciences, Universiti Sains Malaysia, 11800 Pulau Pinang,*
*Malaysia*

*^b Fundamental and Applied Sciences Department, Universiti Teknologi PETRONAS,*
*32610 Perak, Malaysia*

*^c MYCO2 Laboratory Sdn Bhd, Lengkok Kikik 1, Taman Inderawasih, 13600 Pulau*
*Pinang, Malaysia*

**Abstract**

In this study, salting-out assisted liquid-liquid extraction (SALLE) as a simple and efficient
extraction technique followed by high performance liquid chromatography was employed for
the determination of vitamin D3 in milk samples. The sample treatment is based on the use of
water-miscible acetonitrile as the extractant and acetonitrile phase separation under high-salt
conditions. Under the optimum conditions, acetonitrile and ammonium sulfate was used as the
extraction solvent and salting-out agent, respectively. The vitamin D3 extract was separated
using Hypersil ODS (250 x i.d 4.6 mm, 5 µm) high performance liquid chromatography column
that was coupled with diode array detector (HPLC-DAD). Vitamin D2 was used as internal
standard (IS) to offset any variations in chromatographic conditions. The vitamin D3 and the
IS were eluted in 18 minutes. Good linearity ($r^2 > 0.993$) was obtained within the range of
0.025 – 60.0 µg g⁻¹ with the limit of detection (LOD) of 0.015 µg g⁻¹ and limit of quantification
(LOQ) of 0.025 µg g⁻¹. The validated method was applied for the determination of vitamin D3
in milk samples. The recoveries for spiked samples were from 94.40 – 113.50 %.

Introduction

[revised manuscript text omitted]

In accordance to the Hofmeister series, as NH₄⁺ and SO₄²⁻ ions has higher ionic strength
compared to Na⁺ and Cl⁻, thus suggesting that (NH₄)₂SO₄ is more favourable to act as salting-
out reagent. Thus, (NH₄)₂SO₄ was selected for the subsequent study. The effect of different
amounts of (NH₄)₂SO₄ (3.0 – 9.0 g) on the extraction efficiency were also investigated. The
optimum amount of salt was 7.0 g (Figure 1 (d)). The peak area of the extracted vitamin D3
increased with increasing amount of salt up to 7.0 g and the area decreased thereafter.

**Effect of sonication time**

Sonication treatment has two main purposes, i.e., to improve the extraction yields and to
accelerate the phase separation. After the addition of ACN into the sample solution, the effect
of sonication was examined. The extraction yields of vitamin D3 obtained at different times of
sonication (30 and 60 min) showed an increase in peak area as a function of time. Sonication
treatment for 60 min provides good extraction of vitamin D3 from the aqueous phase into the
ACN phase. Sonication treatment more than that should be avoided, because the temperature
of the sample solution started to increase which could cause analyte loss due to excessive
heating as the stability of vitamin D3 was affected by temperature at prolonged time. For
subsequent analysis, 60 min of sonication was chosen.

Adopted extraction conditions

The adopted extraction conditions for 10 mL sample were performed with 3 mL ACN as the extraction solvent and 7.0 g of $(\text{NH}_4)_2\text{SO}_4$ salt as the salting-out reagent with 60 min of sonication.

Method validation

Under the above optimum experimental conditions, the proposed SALLE method was validated in terms of linearity, limits of detection (LOD), limits of quantification (LOQ), repeatability and recovery. A calibration curve was obtained by linear regression of peak area ratio of D3 over internal standard (IS), versus concentration of vitamin D3 ($0.025 - 0.6 \mu\text{g g}^{-1}$). Typical plot is described by

$$y = 3.4634x - 0.0215$$

where y is the ratio peak area of vitamin D3 and IS and x is the concentration of vitamin D3 in $\mu\text{g g}^{-1}$. The coefficient of determination, r^2 obtained is about 0.99, indicating the excellent linearity between ratio of peak areas of vitamin D3 and IS versus concentration of vitamin D3. The LOD and LOQ obtained were 0.015 and $0.025 \mu\text{g g}^{-1}$ respectively. The LOD and LOQ obtained using this method is about similar with previous report using saponification and LLE [31]. This suggest that the SALLE method may be used to analyze milk at $\mu\text{g g}^{-1}$ levels without saponification.

Recovery was evaluated by spiking the milk samples with standard vitamin D3 at different concentrations (Table 1). The recoveries were acceptable (94.4 – 113.5 %). The precision of the analytical procedure was determined using six replicates samples. The repeatability

obtained was good (below than 15 %). It was found that the precision (%RSD) for three
concentration levels (0.05, 0.20 and 0.60 $\mu\text{g g}^{-1}$) were less than 11.45%.

**Analysis of milk samples**

In order to evaluate the applicability of the proposed method, different milk samples were
analyzed and results were compared with the amount claimed by manufacturers (Table 2). All
analyzed samples contained vitamin D3 concentrations that were well as reported by the
manufacturers.

A student paired t- test (at 95% confidence level) shows that there is any significant difference
in the mean values of the manufacturer's claimed values and that of the proposed method (t
stat, 0.082 while t critical, 2.200).

**Comparison to previously reported methods**

Table 3 summarizes the important characteristics of proposed method for the determination of
vitamin D3 in milk samples compared to some previously reported methods. It is readily
apparent that the developed method using SALLE has many advantageous features especially
in terms of solvent consumption, sample preparation time and sensitivity. This proposed
method is simple since there is no saponification method involved, unlike some previously
reported methods. Some of the previously reported methods use SPE, but this requires multi-
steps (equilibration, washing, loading and elution) with single-use SPE cartridges which is
make the experimental longer.

**Conclusions**

Significant advancement is sample pre-treatment method based on SALLE for the
HPLC determination of vitamin D3 in milk is demonstrated. This method is simple, fast

and consumes less amount of solvent compared to previous sample pretreatment
methods which require saponification and elevated temperatures. The extracts are
readily injected for the HPLC analysis, unlike previous methods that require long and
gentle evaporation to remove extracting solvents. It is indeed very interesting if the
SALLE technique can be adopted, with minor modifications, to be applied to the
analysis of other physiologically important metabolites of vitamin D such as 25-OH-D.
If this is possible, it will be a major breakthrough in the analysis of vitamin D
metabolites.

**Ethics.** This article does not present research with any ethical considerations.

**Permission to carry out fieldwork.** This article does not require permission to carry out
fieldwork.

**Data accessibility.** This article does not contain any additional data.

[revised manuscript text omitted]

0.05	94.40 \pm 10.8	11.4
0.20	113.5 \pm 1.78	1.56
0.60	106.6 \pm 4.34	4.07

Table 2: Levels ($\mu\text{g g}^{-1}$) of vitamin D3 in milk samples.

Sample	Claimed ($\mu\text{g g}^{-1}$)	Proposed Method ($\mu\text{g g}^{-1}$) (Average \pmSD)
Milk powder 1	0.042	0.042 \pm 0.034
Milk powder 2	0.094	0.069 \pm 0.001
Milk powder 3	0.066	0.036 \pm 0.038
Milk powder 4	0.080	0.061 \pm 0.015
Milk powder 5	0.050	0.048 \pm 0.005
Milk powder 6	0.080	0.072 \pm 0.016
Milk powder 7	0.250	0.310 \pm 0.028
Liquid Milk 1	0.025	0.029 \pm 0.009
Liquid Milk 2	0.060	0.089 \pm 0.006
Liquid Milk 3	0.025	0.021 \pm 0.000
Liquid Milk 4	0.025	0.020 \pm 0.0016
Liquid Milk 5	0.026	0.018 \pm 0.000

Table 3: Comparison between the proposed SALLE-HPLC method for the determination of vitamin D3 in milk samples and some reported methods.

Samples	Extraction technique	Extraction time (min)	Technique	LOQ ($\mu\text{g g}^{-1}$)	Recovery (%)	Ref
Milk product (solid, liquid)	SALLE	60 min (without saponification)	HPLC-DAD	0.025	94.4 – 113.5	This work
Milk product (solid, liquid)	LLE	70 min saponification + extraction time	LC-UV	0.100	93.0 – 102.0	Staffas & Nyman, (2003)
Infant formula	DLLME	-	LC-MS	0.001	88.0 – 103.0	Vinas et al., (2013)
Infant formula	LLE	10 min saponification + extraction time	LC-MSMS	0.001	93.1 – 110.6	Kwak et al., (2014)
Infant formula	dSPE LLE	Overnight saponification+ 10 min extraction time	HPLC-MSMS	0.005	89.5 – 115.0	Huang et al., (2009)
Milk product (solid)	LLE	Overnight saponification + 20 min extraction time	HPLC-DAD	0.020	73.0 -115.0	Kienen et al., (2008)
Milk product (solid)	Packed fiber SPE	-	HPLC-DAD	0.070	91.7 – 104.5	Chen et al., (2011)
Multi vitamin						
Cod liver fish oil supplement	SPE	-	LC-MS	0.0001	88.8 -99.3	Bartolucci et al., (2011)
Milk product (solid)	LLE	95 min saponification + extraction time	HPLC-UV	0.010	-	Chinese National Standard Method (GB 5413.9—2010)

Appendix B

Reviewer: 1

1. The introduction section is too general. For instance, there is no need to mention in details in which samples vitamin D3 was found. It will be enough to write generally (for e.g. food samples, dietary supplements and biological samples, etc.). But information about SALLE should be included, its principles and benefits. Please, transfer information about SALLE from Result and Discussion section to Introduction one.

I have amended accordingly in text.

2. It is not clear how the conditions of HPLC separation were selected. Was performed any optimization? Information about injection volume, run time, software should be provided.

HPLC parameters and conditions was added in the experimental part.

3. Line 45 p. 4. The word "extraction" is repeated 2 times . I would recommend to change to "SALLE procedure" or "Extraction procedure".

I have amended accordingly in text.

4. The description of extraction procedure is not clear. How was performed experiment? What tools were used?

Amendments have been made to provide clearer description on the procedure. The extraction was carried out with basic apparatus in the lab.

5. Line 59, p. 4. It is not clear how much of vitamin C was added. Why was it necessary to prepare vitamin C in ethanol?

- The preparation of vitamin C in ethanol was followed as mentioned in AOAC method 2002.05 (Determination of Cholecalciferol (Vitamin D3) in Selected Food).

6. Line 6, p. 5. Why was added formic acid?

- Formic acid was applied in order to enhance pre-treatment for precipitation of the protein, to separate fats and any water-soluble components combined with vitamin D3 extraction.

7. In Result and Discussion section should be provided only obtained results. Line 22-46, there is known information, which can be transferred to introduction section.

I have amended accordingly in text.

8. Preliminary study is not clear. The results of this study are not provided, just only mention the reference.

- The preliminary results obtained from the experiments were very similar to the one reported in the reference. Hence, the authors agreed to mentioned the results without providing figure.

9. Line 15, 38 p.7. The titles of these items can be change to "Effect of type of extraction solvent", "Effect of extraction solvent volume".

I have amended accordingly in text.

10. Line 17-33 p. 7. A lot of a well-known information, but this is not the results of this experiment. Here should be provided only the results of the test.

I have amended accordingly in text.

11. Line 26, p. 7. At the beginning of Results and discussion section it was mention about

magnesium sulfate, ammonium acetate (but, not only these salts can be used for phase separation), after the effect of NaCl and ammonium sulfate was studied (line 6, p 8).
I have amended accordingly in text.

12. The coefficient of determination should contain 4 digits (line 39, p.9).
I have amended accordingly in text.

13. Please provide recovery and RSD to Table 2. From my point of view, the developed method is not suitable for analysis milk powder 1 and 3 (from your data, RSD in this sample is about 81%), for milk powder 3 RSD is 105.5%). How can you explain these results of analysis? What did effect on such critical deviation? In Table 2, I would to recommend to write results in ng/g. The way of writing of SD as 0.000 is not acceptable (liquid milk 3, liquid milk 4).

Amendments was made in Table 2 where the RSD values were added. Milk powder 1 and 3 was omitted.

14. In section Comparison to previously reported methods, referenced should be added and in Table 2 references should be modified according to the same references style like in the whole text.

Table 3 was amended accordingly.

15. To the capture of Fig 1 should be added information about extraction conditions. Fig 3b,c (ml and g should be mention only one time (on the scale or in the title of the axis).

The caption for Figure 1 was added in the text.

Appendix C**ROYAL SOCIETY
OPEN SCIENCE****Salting-out assisted liquid-liquid extraction coupled with
high performance liquid chromatography for determination
of vitamin D3 in milk samples**

Journal:	Royal Society Open Science
Manuscript ID	RSOS-182170
Article Type:	Research
Date Submitted by the Author:	10-Feb-2019
Complete List of Authors:	Sazali, Nur Hidayah; Universiti Sains Malaysia, School of Chemical Sciences Al-shishani, Anas Saad, Bahruddin; Universiti Teknologi PETRONAS, Fundamental & Applied Sciences Chong, Moi Me; MYCO2 Laboratory Sdn Bhd Chew, Ker Yin; MYCO2 Laboratory Sdn Bhd Miskam, Mazidatulakman; Universiti Sains Malaysia, School of Chemical Sciences
Subject:	Analytical chemistry < CHEMISTRY, Green chemistry < CHEMISTRY
Keywords:	Salting-out assisted liquid-liquid extraction, vitamin D3, High performance liquid chromatography, Milk samples
Subject Category:	Chemistry

**Salting-out assisted liquid-liquid extraction coupled with high performance liquid**
**chromatography for determination of vitamin D3 in milk samples**

Nur Hidayah SAZALI^a, Anas ALSHISHANI^a, Bahruddin SAAD^b, Ker Yin CHEW^c, Moi Me
CHONG^c, Mazidatulakmam MISKAM^{*a}

*^a School of Chemical Sciences, Universiti Sains Malaysia, 11800 Pulau Pinang,*
*Malaysia*

*^b Fundamental and Applied Sciences Department, Universiti Teknologi PETRONAS,*
*32610 Perak, Malaysia*

*^c MYCO2 Laboratory Sdn Bhd, Lengkok Kikik 1, Taman Inderawasih, 13600 Pulau*
*Pinang, Malaysia*

**Abstract**

In this study, salting-out assisted liquid-liquid extraction (SALLE) as a simple and efficient
extraction technique followed by high performance liquid chromatography was employed for
the determination of vitamin D3 in milk samples. The sample treatment is based on the use of
water-miscible acetonitrile as the extractant and acetonitrile phase separation under high-salt
conditions. Under the optimum conditions, acetonitrile and ammonium sulfate was used as the
extraction solvent and salting-out agent, respectively. The vitamin D3 extract was separated
using Hypersil ODS (250 x i.d 4.6 mm, 5 µm) high performance liquid chromatography column
that was coupled with diode array detector (HPLC-DAD). Vitamin D2 was used as internal
standard (IS) to offset any variations in chromatographic conditions. The vitamin D3 and the
IS were eluted in 18 minutes. Good linearity ($r^2 > 0.993$) was obtained within the range of
0.025 – 60.0 µg g⁻¹ with the limit of detection (LOD) of 0.015 µg g⁻¹ and limit of quantification
(LOQ) of 0.025 µg g⁻¹. The validated method was applied for the determination of vitamin D3
in milk samples. The recoveries for spiked samples were from 94.40 – 113.50 %.

Introduction

Cholecalciferol, commonly known as vitamin D₃ is a fat-soluble vitamin, is of great nutritional interest that support metabolism processes and improve the efficiency of proteins and enzymes [1]. It plays important roles within the body as it regulates blood calcium and phosphorus levels by promoting the absorption in intestines and reabsorption of calcium in kidney that is key to the mineralization of the bones [2,3]. Vitamin D deficiency is recognized as one of most common mild chronic medical conditions in the world. It can lead to soft, thin and brittle bones, a disease known as rickets in children [4,5].

Vitamin D₃ is naturally found in trace amounts in some foods such as some fatty fishes (mackerel, salmon, sardines), fish liver oils, and eggs from hens that have been fed vitamin D₃. Among dairy products, processed milk products and infant formulas are fortified with vitamin D₃. However, its content is low, ranging from 4 to 40 IU (0.1–1.0 µg/L) and which corresponds mainly to vitamin D₃ and 25-hydroxy-vitamin D₃ [6]. Vitamin D₃ is also sensitive to heat and light and is easily oxidized [7].

[revised manuscript text omitted]

In accordance to the Hofmeister series, as NH₄⁺ and SO₄²⁻ ions has higher ionic strength
compared to Na⁺ and Cl⁻, thus suggesting that (NH₄)₂SO₄ is more favourable to act as salting-
out reagent. Thus, (NH₄)₂SO₄ was selected for the subsequent study. The effect of different
amounts of (NH₄)₂SO₄ (3.0 – 9.0 g) on the extraction efficiency were also investigated. The
optimum amount of salt was 7.0 g (Figure 1 (d)). The peak area of the extracted vitamin D3
increased with increasing amount of salt up to 7.0 g and the area decreased thereafter.

**Effect of sonication time**

Sonication treatment has two main purposes, i.e., to improve the extraction yields and to
accelerate the phase separation. After the addition of ACN into the sample solution, the effect
of sonication was examined. The extraction yields of vitamin D3 obtained at different times of
sonication (30 and 60 min) showed an increase in peak area as a function of time. Sonication
treatment for 60 min provides good extraction of vitamin D3 from the aqueous phase into the
ACN phase. Sonication treatment more than that should be avoided, because the temperature
of the sample solution started to increase which could cause analyte loss due to excessive
heating as the stability of vitamin D3 was affected by temperature at prolonged time. For
subsequent analysis, 60 min of sonication was chosen.

Adopted extraction conditions

The adopted extraction conditions for 10 mL sample were performed with 3 mL ACN as the extraction solvent and 7.0 g of $(\text{NH}_4)_2\text{SO}_4$ salt as the salting-out reagent with 60 min of sonication.

Method validation

Under the above optimum experimental conditions, the proposed SALLE method was validated in terms of linearity, limits of detection (LOD), limits of quantification (LOQ), repeatability and recovery.  calibration curve was obtained by linear regression of peak area ratio of D3 over internal standard (IS), versus concentration of vitamin D3 ($0.025 - 0.6 \mu\text{g g}^{-1}$). Typical plot is described by

$$y = 3.4634x - 0.0215$$

 where y is the ratio peak area of vitamin D3 and IS and x is the concentration of vitamin D3 in $\mu\text{g g}^{-1}$. The coefficient of determination, r^2 obtained is about 0.99 indicating the excellent linearity between ratio of peak areas of vitamin D3 and IS versus concentration of vitamin D3. The LOD and LOQ obtained were 0.015 and $0.025 \mu\text{g g}^{-1}$ respectively. The LOD and LOQ obtained using this method is about similar with previous report using saponification and LLE [31]. This suggest that the SALLE method may be used to analyze milk at $\mu\text{g g}^{-1}$ levels without saponification.

Recovery was evaluated by spiking the milk samples with standard vitamin D3 at different concentrations (Table 1). The recoveries were acceptable (94.4 – 113.5 %). The precision of the analytical procedure was determined using six replicates samples. The repeatability

obtained was good (below than 15 %). It was found that the precision (%RSD) for three
concentration levels (0.05, 0.20 and 0.60 $\mu\text{g g}^{-1}$) were less than 11.45%.

**Analysis of milk samples**

In order to evaluate the applicability of the proposed method, different milk samples were
analyzed and results were compared with the amount claimed by manufacturers (Table 2). All
analyzed samples contained vitamin D3 concentrations that were well as reported by the
manufacturers.

A student paired t- test (at 95% confidence level) shows that there is any significant difference
in the mean values of the manufacturer's claimed values and that of the proposed method (t
stat, 0.082 while t critical, 2.200).

**Comparison to previously reported methods**

Table 3 summarizes the important characteristics of proposed method for the determination of
vitamin D3 in milk samples compared to some previously reported methods. It is readily
apparent that the developed method using SALLE has many advantageous features especially
in terms of solvent consumption, sample preparation time and sensitivity. This proposed
method is simple since there is no saponification method involved, unlike some previously
reported methods. Some of the previously reported methods use SPE, but this requires multi-
steps (equilibration, washing, loading and elution) with single-use SPE cartridges which is
make the experimental longer.

**Conclusions**

Significant advancement is sample pre-treatment method based on SALLE for the
HPLC determination of vitamin D3 in milk is demonstrated. This method is simple, fast

and consumes less amount of solvent compared to previous sample pretreatment
methods which require saponification and elevated temperatures. The extracts are
readily injected for the HPLC analysis, unlike previous methods that require long and
gentle evaporation to remove extracting solvents. It is indeed very interesting if the
SALLE technique can be adopted, with minor modifications, to be applied to the
analysis of other physiologically important metabolites of vitamin D such as 25-OH-D.
If this is possible, it will be a major breakthrough in the analysis of vitamin D
metabolites.

**Ethics.** This article does not present research with any ethical considerations.

**Permission to carry out fieldwork.** This article does not require permission to carry out
fieldwork.

**Data accessibility.** This article does not contain any additional data.

[revised manuscript text omitted]

0.05	94.40 \pm 10.8	11.4
0.20	113.5 \pm 1.78	1.56
0.60	106.6 \pm 4.34	4.07

Table 2: Levels ($\mu\text{g g}^{-1}$) of vitamin D3 in milk samples.

Sample	Claimed ($\mu\text{g g}^{-1}$)	Proposed Method ($\mu\text{g g}^{-1}$) (Average \pmSD)
Milk powder 1	0.042	0.042 \pm 0.034
Milk powder 2	0.094	0.069 \pm 0.001
Milk powder 3	0.066	0.036 \pm 0.038
Milk powder 4	0.080	0.061 \pm 0.015
Milk powder 5	0.050	0.048 \pm 0.005
Milk powder 6	0.080	0.072 \pm 0.016
Milk powder 7	0.250	0.310 \pm 0.028
Liquid Milk 1	0.025	0.029 \pm 0.009
Liquid Milk 2	0.060	0.089 \pm 0.006
Liquid Milk 3	0.025	0.021 \pm 0.000
Liquid Milk 4	0.025	0.020 \pm 0.0016
Liquid Milk 5	0.026	0.018 \pm 0.000

Table 3: Comparison between the proposed SALLE-HPLC method for the determination of vitamin D3 in milk samples and some reported methods.

Samples	Extraction technique	Extraction time (min)	Technique	LOQ ($\mu\text{g g}^{-1}$)	Recovery (%)	Ref
Milk product (solid, liquid)	SALLE	60 min (without saponification)	HPLC-DAD	0.025	94.4 – 113.5	This work
Milk product (solid, liquid)	LLE	70 min saponification + extraction time	LC-UV	0.100	93.0 – 102.0	Staffas & Nyman, (2003)
Infant formula	DLLME	-	LC-MS	0.001	88.0 – 103.0	Vinas et al., (2013)
Infant formula	LLE	10 min saponification + extraction time	LC-MSMS	0.001	93.1 – 110.6	Kwak et al., (2014)
Infant formula	dSPE LLE	Overnight saponification+ 10 min extraction time	HPLC-MSMS	0.005	89.5 – 115.0	Huang et al., (2009)
Milk product (solid)	LLE	Overnight saponification + 20 min extraction time	HPLC-DAD	0.020	73.0 -115.0	Kienen et al., (2008)
Milk product (solid)	Packed fiber SPE	-	HPLC-DAD	0.070	91.7 – 104.5	Chen et al., (2011)
Multi vitamin						
Cod liver fish oil supplement	SPE	-	LC-MS	0.0001	88.8 -99.3	Bartolucci et al., (2011)
Milk product (solid)	LLE	95 min saponification + extraction time	HPLC-UV	0.010	-	Chinese National Standard Method (GB 5413.9—2010)

Appendix D

12th July 2019.

Dr. Mazidatulakmam Miskam,
School of Chemical Sciences,
Universiti Sains Malaysia,
11800 USM Pulau Pinang,
Malaysia.

Email: mazidatul@usm.my

Dear,

Editorial Board of Royal Society Open Science

Enclosed is the revised manuscript “**Salting-out assisted liquid-liquid extraction coupled with high performance liquid chromatography for determination of vitamin D3 in milk samples**” by N.H Sazali et al, with reference to the comments by Reviewer #1 and Reviewer #2 whom we would like to thank for his/her time and guidance in helping us prepare a good scientific manuscript.

2. We have addressed each comment individually and provided a detailed respond to reviewer 1. New reference was added in page 5 to address the reviewer concern on the concentration and significant figures were reduced to 1 to avoid excess figure number in Table 2. We hope the answers provided addresses the comments given satisfactorily.

3. We sincerely believe that the readers of Royal Society Open Science would find our work to be both interesting and useful.

Thank you.

Yours Sincerely,
Dr. Mazidatulakmam Miskam
Universiti Sains Malaysia, Malaysia.